# Panorama of Two Decades of Maternal Deaths in Brazil: Retrospective Ecological Time Series

**DOI:** 10.3390/nursrep15110396

**Published:** 2025-11-11

**Authors:** Gustavo Gonçalves dos Santos, Giovana Aparecida Gonçalves Vidotti, Carolliny Rossi de Faria Ichikawa, Cindy Ferreira Lima, Leticia de Almeida Dionizio, Janize Silva Maia, Karina Franco Zihlmann, Joaquim Guerra de Oliveira Neto, Wágnar Silva Morais Nascimento, Alexandrina Maria Ramos Cardoso, Júlia Maria das Neves Carvalho, Patrícia Lima Ferreira Santa Rosa, Ricardo José Oliveira Mouta, Cesar Henrique Rodrigues Reis, Cláudia de Azevedo Aguiar, Débora de Souza Santos, Bruno Pereira da Silva, Anderson Lima Cordeiro da Silva, Edson Silva do Nascimento, Beatriz Maria Bermejo Gil, Leticia López Pedraza

**Affiliations:** 1Escola Paulista de Medicina, Universidade Federal de São Paulo (EPM/UNIFESP), Edifício Octávio de Carvalho—Rua Botucatu, 740—5º Andar, Sala 508, São Paulo 04023-062, SP, Brazil; goncalves.giovana2@gmail.com; 2Faculdade de Medicina, Centro Universitário da Faculdade de Medicina do ABC (FMABC), Santo André 09060-870, SP, Brazil; carolliny.ichikawa@fmabc.net; 3Faculdade de Enfermagem, Universidade Santo Amaro, Campus Interlagos (UNISA), São Paulo 04829-300, SP, Brazil; cindy.lima@alumni.usp.br (C.F.L.); ldionizio@prof.unisa.br (L.d.A.D.); 4Faculdade de Enfermagem, Centro Universitário da Faculdade de Engenharia de Sorocaba (FACENS), Sorocaba 18085-784, SP, Brazil; janizecs@yahoo.com.br; 5Departamento de Saúde, Educação e Sociedade, Universidade Federal de São Paulo, Campus Baixada Santista (UNIFESP/BS), Santos 11015-020, SP, Brazil; karina.zihlmann@unifesp.br; 6Faculdade de Ciências da Saúde, Universidade Federal do Norte do Tocantins (UFNT/FCS), Araguaína 77814-350, TO, Brazil; joaquim.neto@ufnt.edu.br; 7Centro de Ciências da Saúde, Universidade Federal do Piauí (CCS/UFPI), Teresina 64049-550, PI, Brazil; wagnarnascimento@gmail.com; 8Escola Superior de Enfermagem do Porto (ESEP), 4200-072 Porto, PT, Portugal; alex@esenf.pt; 9Escola Superior de Enfermagem de Coimbra (ESEnfC), 3046-851 Coimbra, PT, Portugal; juliacarvalho@esenfc.pt; 10Escola de Enfermagem de Ribeirão Preto, Universidade de São Paulo (EERP/USP), Ribeirão Preto 14040-902, SP, Brazil; patriciasantarosa@usp.br; 11Faculdade de Enfermagem, Universidade do Estado do Rio de Janeiro (FACENF/UERJ), Rio de Janeiro 20551-030, RJ, Brazil; ricardomoutta@gmail.com; 12Programa de Pós-graduação em Enfermagem, Universidade Federal de Mato Grosso do Sul (UFMS), Três Lagoas 79070-900, MS, Brazil; cesar.h@ufms.br; 13Instituto de Ciências da Saúde, Universidade Federal do Triângulo Mineiro (UFTM/ICS), Uberaba 38025-180, MG, Brazil; claudia.aguiar@uftm.edu.br; 14Faculdade de Enfermagem, Universidade Estadual de Campinas (UNICAMP), Campinas 13000-000, SP, Brazil; deborass@unicamp.br; 15Centro Multidisciplinar, Universidade Federal do Acre (UFAC), Cruzeiro do Sul 69980-000, AC, Brazil; bruno.silva@ufac.br; 16Escola de Filosofia, Letras e Ciências Humanas, Universidade Federal de São Paulo (EFLCH/Unifesp), Guarulhos 07252-312, SP, Brazil; 17Faculdade de Medicina, Universidade de São Paulo (FMUSP), São Paulo 01246-903, SP, Brazil; enfandersoncordeiro@gmail.com (A.L.C.d.S.); enfedsonnascimento@gmail.com (E.S.d.N.); 18Facultad de Enfermería y Fisioterapia, Universidad de Salamanca (USAL), 37007 Salamanca, ES, Spain; beatriz.bermejo@usal.es; 19Facultad de Enfermería, Fisioterapia y Podología, Universidad Complutense de Madrid (UCM), 28040 Madrid, ES, Spain; leticia.lopez@cruzroja.es

**Keywords:** women’s health, sexual and reproductive health, pregnancy, postpartum period, maternal mortality, health information system

## Abstract

**Background:** Maternal mortality remains a significant public health challenge in Brazil, reflecting persistent social, racial, and regional inequalities. **Objectives:** This study aimed to analyze trends and characteristics of maternal deaths in Brazil from 2000 to 2020, identifying sociodemographic and clinical risk factors. **Methods:** A retrospective, population-based study was conducted using secondary data from the Sistema de Informações sobre Mortalidade (SIM) of the Brazilian Ministry of Health. Descriptive and comparative statistical analyses were performed, including chi-square (χ^2^) goodness-of-fit tests and 95% confidence intervals calculated by the Wilson method. Variables were stratified by region, age, race, education, type and period of death, and investigation status. **Results:** A total of 40,907 maternal deaths were recorded. From 2000 to 2020, Brazil recorded 40,907 maternal deaths. The maternal mortality ratio decreased from approximately 74 deaths per 100,000 live births in 2000 to 57 per 100,000 in 2020, representing a 23% reduction over two decades. Most deaths occurred among young, Brown women with low education levels, particularly in the Southeast and Northeast regions. Direct obstetric causes accounted for two-thirds of cases, and most deaths occurred in hospital settings, mainly during the early puerperium. **Conclusions:** Despite progress, maternal mortality in Brazil remains above the WHO target and is strongly influenced by social inequities. Strengthening primary care, improving referral networks, expanding postpartum follow-up, and enhancing surveillance systems are essential for preventing avoidable deaths and achieving reproductive justice.

## 1. Introduction

Maternal mortality is defined as the death of a woman during pregnancy, childbirth or up to 42 days after the end of pregnancy, due to causes related to pregnancy or its management, excluding accidental or incidental causes. It is an important indicator of the quality of health care provided to pregnant and postpartum women and is used globally as a parameter to assess the effectiveness of health systems [1,2].

The main causes of maternal mortality in Brazil include hypertensive disorders during pregnancy (such as pre-eclampsia and eclampsia), hemorrhages, puerperal infections and complications resulting from unsafe abortions. These causes are, in most cases, preventable with adequate and timely care. Although Brazil has recorded a reduction in the maternal mortality ratio (MMR) in recent decades, the figures are still above the parameters recommended by the World Health Organization (WHO), highlighting the persistence of inequalities in access to and quality of obstetric care [3,4].

The country exhibits marked regional and socioeconomic disparities in maternal health, with higher MMRs in the North and Northeast regions, reflecting differences in service availability and professional training. Socio-demographic factors such as maternal age, education, and race also influence outcomes [5,6]. Despite improvements in surveillance, underreporting of maternal deaths persists. In 2019, a correction factor of 1.05% was proposed, indicating that official figures slightly underestimate the real magnitude of mortality [7]. Near-miss events, cases in which women survive severe complications, are also important for identifying risks and strengthening care practices [8].

Many deaths occur in the early postpartum period, often after hospital discharge, revealing failures in continuity of care [9]. The COVID-19 pandemic intensified this scenario [10]. During the COVID-19 pandemic, Brazil recorded a significant increase in maternal mortality. In 2021, the MMR reached 10,753 deaths per 100,000 live births, almost double the rate of 5531 recorded in 2019. In 2022, there was a reduction in the MMR to 57 deaths per 100,000 live births, returning to levels similar to those of the pre-pandemic period. Despite this improvement, Brazil is still far off the WHO target, which predicts an MMR of less than 30 deaths per 100,000 live births by 2030. In addition, large regional disparities persist; in 2023, the North region had the highest rate, while the South region had the lowest rate [11]. The data show that social and institutional factors contribute to precarious care, especially in obstetric emergency situations [12], with recurring failures in the surveillance and reporting of maternal deaths [13].

Given this scenario, the present investigation sought to answer the following research question: What are the trends and main sociodemographic, clinical, and regional factors associated with maternal mortality in Brazil between 2000 and 2020? Unlike previous Ministry of Health surveillance reports and evaluations from the Rede Cegonha, which mainly presented descriptive indicators and institutional performance data for shorter periods, this study integrates two decades of population-based data and applies comparative analyses across social, racial, and regional dimensions. By combining sociodemographic, clinical, and geographic information, it provides a comprehensive overview of persistent inequities and identifies priority areas for intervention at the national level. This population-based approach fills an important gap in the literature by connecting temporal trends to social determinants of health, offering insights beyond surveillance summaries. Nurses and midwives are key frontline professionals in identifying risks, managing obstetric emergencies, and ensuring postnatal follow-up. By identifying the critical moments and groups most affected by maternal deaths, this research supports targeted improvements in care coordination that can strengthen the Brazilian maternal health system.

The objective of this study was to analyze the temporal evolution and factors associated with maternal mortality in Brazil over two decades, with an emphasis on sociodemographic, clinical and regional characteristics.

## 2. Materials and Methods

### 2.1. Study Design

This is a retrospective ecological time-series analysis [14,15].

### 2.2. Population and Period

The study population consisted of all registered cases of maternal death among women aged 10 to 49 years in Brazil, recorded between 1 January 2000 and 31 December 2020, which were included in this study. Following the WHO definition, a maternal death was considered eligible if it met at least one of the following criteria: death during pregnancy, childbirth, or abortion; death in the early puerperium, up to 42 days after the end of pregnancy; or death in the late puerperium, from 43 days to less than one year after the end of pregnancy. Women were included in the analysis if their death record fulfilled any one of these conditions, regardless of the underlying obstetric cause classification.

During the period analyzed (2000 to 2020), the Sistema de Informações sobre Mortalidade (SIM) recorded a total of 204,574,000 deaths among women in Brazil. Of these, 40,907 were classified as maternal deaths, representing approximately 0.02% of the total female deaths in the country during the period. Among these maternal deaths, 40,016 (97.8%) were directly related to obstetric causes classified in Chapter XV of ICD-10, referring to pregnancy, childbirth, and the puerperium.

The study design followed the principles of observational epidemiological research, using secondary data obtained from a national health surveillance system. The Brazilian territory, composed of 26 states and the Federal District, is geographically divided into five macro-regions (North, Northeast, Southeast, South, and Central-West), which were used for stratified analysis. In total, 40,907 maternal deaths recorded in Brazil between 2000 and 2020 were analyzed. Of these, 40,016 (97.8%) were classified as related to Chapter XV of ICD-10—Pregnancy, Childbirth and Puerperium. This number represents all maternal deaths directly attributable to obstetric causes during the period studied.

### 2.3. Data Sources and Variables

Data was extracted from the SIM of the Brazilian Ministry of Health (https://opendatasus.saude.gov.br/dataset/sim, access date 10 June 2025). The system aggregates data from death certificates collected nationwide and includes information on demographics, causes of death, location of death, investigation status and clinical circumstances. The main variables analyzed were (1) sociodemographic characteristics, age, race, marital status, and level of education; (2) geographic region, cause of death, classified according to the ICD-10 chapters and grouped into direct obstetric, indirect obstetric and unspecified causes; (3) period of occurrence of death, outside of pregnancy/puerperium, inconsistent or unknown periods; (4) place of death, hospital, home, public road, other establishments, or unknown; and (5) investigation status, investigated with or without a medical record, not investigated or not applicable.

The variable referring to ethnic–racial classification was obtained from the database used, where it is recorded as “race.” This classification follows the criteria standardized by the Instituto Brasileiro de Geografia e Estatística (IBGE), which considers the following categories: white, Black, Brown, Asian and indigenous. It should be noted that this term does not imply any judgment or hierarchy between groups and is used exclusively for epidemiological purposes. For this study, the categories were analyzed as recorded in the database, ensuring methodological consistency and transparency in the interpretation of the results.

Records with missing or “ignored” information on key sociodemographic variables (education, race, marital status) were retained in the analyses as separate categories to avoid data loss and bias. The proportions of missing data were reported in the descriptive tables. Sensitivity analyses were performed to ensure that the inclusion of these categories did not substantially alter the distribution of other variables.

### 2.4. Data Management and Statistical Analysis

Data was downloaded in Comma-Separated Values (CSVs) format and processed using R software, version 4.3.1. Analyses were conducted with the epiR (version 2.0.63) and stats (base package) libraries. Graphical visualizations were generated using ggplot2 (version 3.5.0).

Descriptive results were expressed as absolute and relative frequencies with 95% CIs calculated by the Wilson method. Descriptive statistics were applied to characterize the population. Absolute and relative frequencies were calculated for all categorical variables. The MMR was computed per 100,000 live births using birth data from Sistema de Informações sobre Nascidos Vivos (SINASC) as the denominator. To assess differences in the distribution of deaths by categories, Chi-square (χ^2^) goodness-of-fit tests were applied, comparing observed versus expected frequencies under the null hypothesis of equal distribution among categories. For each category, 95% confidence intervals (95% CI) were calculated for proportions using the Wilson method. Analyses were performed with the support of the R epiR and stats packages. Data on maternal deaths were organized into thematic categories according to the chapters of the International Classification of Diseases—10th Revision (ICD-10), type of cause (direct, indirect, or unspecified obstetric), period of death (pregnancy, childbirth, early or late puerperium, among others), place of occurrence (hospital, home, public road, etc.), and investigation status (with or without documentation, not investigated, not applicable).

Data validation procedures were implemented to ensure consistency and reliability. Records were checked for internal consistency using ICD-10 coding and variable cross-validation (e.g., period of death versus cause). Duplicate records were identified through combinations of death certificate number, municipality code, and year, and subsequently excluded. Missing or “ignored” categories for key variables (education, race, marital status) were retained as explicit categories to prevent loss of information, and their proportions were reported in descriptive tables. Sensitivity analyses confirmed that the inclusion of these categories did not substantially alter the distribution of the remaining variables.

### 2.5. Ethical Considerations

The study used publicly available and fully anonymized secondary data from the SIM. According to the Brazilian National Health Council Resolution No. 510/2016, studies based exclusively on secondary data that do not allow individual identification are exempt from Research Ethics Committee approval. All datasets are open-access and can be retrieved from the Ministry of Health’s public data repository. The authors confirm full adherence to the STROBE guidelines, ensuring transparency and completeness in the reporting of observational data. All datasets used are publicly accessible through the Brazilian Ministry of Health’s information systems, allowing for complete data reproducibility. The analytical procedures were performed using R software (version 4.3.1).

## 3. Results

A total of 40,907 maternal deaths were analyzed. Descriptive statistics summarize sociodemographic and clinical profiles, followed by a temporal overview of maternal mortality in Brazil over the past two decades. Table 1 and Table 2 summarize the sociodemographic and clinical distribution of maternal deaths in Brazil from 2000 to 2020. The highest proportions were recorded in the Southeast (34.5%) and Northeast (33.6%) regions, among women aged 20–39 years (78%), self-identified as Brown (48%), single (50%), and with 4–11 years of schooling (53%). Direct obstetric causes accounted for 67.5% of deaths, mainly during the early puerperium (47%) and predominantly in hospital settings (91%).

Figure 1 illustrates the annual trend in the maternal mortality ratio (MMR) in Brazil from 2000 to 2020. Over two decades, the MMR decreased from approximately 74 deaths per 100,000 live births in 2000 to 57 in 2020, corresponding to a 23% reduction. Despite this decline, the curve remained above the WHO target (<30 deaths/100,000) throughout the period, showing minor fluctuations after 2015.

Direct obstetric causes were responsible for the majority of deaths, while indirect causes accounted for nearly one-third. A smaller proportion remained classified as unspecified. The predominance of direct causes indicates that most maternal deaths in Brazil are still preventable through timely and qualified obstetric care (Figure 2).

Regarding the timing of deaths, the early puerperium was the most critical period, followed by pregnancy and childbirth or abortion. A smaller proportion occurred during the late puerperium, while deaths outside pregnancy or the puerperal period were rare (Figure 3).

This distribution reinforces that the postpartum period, especially the first six weeks after delivery, represents the most vulnerable phase for maternal health. Most maternal deaths occurred in hospital settings, with a small number recorded at home or in other locations. Although hospital care predominated, the persistence of fatal outcomes within institutional settings suggests gaps in the quality of emergency obstetric care, early diagnosis, and continuity of postpartum follow-up. Overall, the findings reveal a consistent pattern of maternal deaths over two decades, with predominance of preventable causes and a clear association with social and regional inequities.

## 4. Discussion

### 4.1. Regional Inequalities

Over the past two decades, Brazil has achieved only modest reductions in maternal mortality, revealing persistent regional and social disparities that reflect the country’s broader structural inequalities. The North and Northeast regions continue to have the highest maternal mortality rates, while the South and Southeast maintain lower and more stable rates. This territorial gradient reflects the unequal distribution of health infrastructure, professional training, and socioeconomic development [5,6].

Recent spatial and time-series analyses confirm that maternal mortality is spatially concentrated in states such as Amazonas, Maranhão, Bahia, and Pará, where poverty, geographic isolation, and a shortage of specialized professionals converge to increase preventable deaths [5]. The Fundação Amazônia de Amparo a Estudos e Pesquisas (FAPESPA) report (2024) further reinforces that, in Pará, maternal mortality remained above 90 per 100,000 live births between 2019 and 2023, far exceeding the national average [11]. Beyond geography, racial and socioeconomic inequalities exert a cumulative effect on maternal outcomes. Black women have significantly higher MMR compared to White women, even after adjusting for education and region, highlighting the pervasive impact of structural racism on access to and quality of obstetric care [16,17]. Similarly, Leal et al. (2017) documented that Black and Brown women more frequently face disrespect, delayed care, and denial of pain relief during labor, demonstrating institutionalized racial bias in maternity care [10].

The literature also highlights how racial and socioeconomic inequalities intersect with regional disparities. Studies indicate that Black and Brown women are up to 2.5 times more likely to die from maternal causes than White women, largely due to discriminatory practices, barriers to prenatal care, and poor-quality hospital services [10,12,16]. Recent studies corroborate this finding, showing that factors such as race, education, and geographic location are strongly associated with the risk of maternal death. Black and brown women have a 2.5 times greater risk of dying from maternal causes compared to white women, and this risk is higher in regions with lower coverage and quality of obstetric services [16].

### 4.2. Risk Factors and Obstetric Causes

The preventable deaths of millions of women each decade are not only due to biomedical complications of pregnancy, childbirth, and the postpartum period, but are also tangible manifestations of the prevailing determinants of maternal health and persistent inequalities in global health and socioeconomic development [17]. Brazil, with an estimated MMR of 107.53/100,000 in 2021, is above the target established by the Sustainable Development Goals (SDGs), highlighting the need for more effective and equitable policies [18].

Compared to other middle-income countries, Brazilian data are intermediate. A recent study found that the incidence of hypertensive disorders during pregnancy, the leading direct cause of maternal death, is lower in Brazil than in Russia and India, but still higher than levels observed in China and South Africa [19,20]. This reveals the positive impact of some Brazilian public policies on maternal health, such as the Stork Network Strategy, but also the limitations in the coverage and resolution of obstetric care in vulnerable regions [21].

The predominance of direct causes of maternal mortality, especially those related to Chapter XV of ICD-10, was also evident in this study. Causes such as postpartum hemorrhage, hypertensive disorders, and puerperal infections remain among the main lethal events, despite being widely recognized as preventable. The lack of early diagnosis and timely interventions during prenatal care and childbirth remains a determining factor in negative outcomes [22]. The high occurrence of these deaths in hospital settings, evidenced in our analysis, suggests serious care failures in the health system, highlighting the need to strengthen the training of obstetric care teams and the organization of care networks [23]. In addition to direct causes, such as hemorrhage, infections, and hypertensive syndromes, the proportion of deaths associated with indirect causes, such as cardiovascular disease and mental disorders, has been increasing, particularly in the late postpartum period [24,25]. In the United Kingdom, for example, the leading causes of maternal death in 2022 were suicide and heart problems. This suggests the need to expand postpartum monitoring and integrate mental health and primary care interventions with maternal health [26].

Maternal mortality in the immediate postpartum period (up to 42 days after delivery) accounted for almost 10% of all deaths analyzed, corroborating the literature that identifies this period as critical for the occurrence of serious complications. Women who do not have access to continuous monitoring after hospital discharge are more likely to develop untreated complications, such as infections, thrombosis, and mental disorders, which increases the risk of preventable death [27].

### 4.3. Health System Factors

Another relevant point concerns the low completeness of maternal death investigations. In our study, more than 13% of records were not investigated or were investigated incompletely, which compromises the quality of surveillance and hinders the identification of systemic failures. In this sense, the lack of active maternal mortality committees in some Brazilian regions hinders the development of corrective actions and impedes institutional learning from adverse events [28]. Investing in more robust reporting systems and teams dedicated to the critical analysis of deaths is essential to breaking this cycle of invisibility and neglect [29].

Over the past two decades, Brazil has achieved only modest reductions in maternal mortality, reflecting persistent social and territorial inequalities. The findings reinforce the importance of strengthening public health strategies such as the Rede Cegonha and the National Policy for Comprehensive Women’s Health Care (PNAISM). Both initiatives aimed to expand prenatal coverage, qualify childbirth care, and integrate postpartum follow-up, but their implementation has been uneven across regions and social groups [21,22].

A study conducted with primary care professionals revealed stigmatizing perceptions and misinformation about the reproductive rights of trans men, indicating the urgent need for ethical, technical, and political training for health teams. The effective inclusion of these populations in prenatal care and surveillance strategies is essential not only to reduce clinical risks but also to ensure reproductive justice, as advocated by the SDGs. Ignoring these experiences can contribute to the underreporting of deaths and the perpetuation of an exclusionary health system, which compromises gender equity and the right to comprehensive health [30,31].

Strengthening Maternal Mortality Surveillance Committees is also essential. The proportion of uninvestigated deaths in our data (approximately 13%) reflects weaknesses in local committee operations and information flow between hospitals and municipal health departments. Previous evaluations by the Ministry of Health have emphasized that incomplete investigations hinder corrective actions and mask preventable causes [1]. These findings suggest that policies must go beyond expanding access, focusing instead on the quality, continuity, and territorial equity of care. Integrating Rede Cegonha’s clinical protocols into primary care, improving referral networks, and ensuring postpartum follow-up, particularly within 42 days after delivery, are key measures to address the preventable nature of most maternal deaths identified in this study [23,24,31].

### 4.4. Nursing Implications

The results of this study echo other national findings, reinforcing persistent regional and racial inequalities in Brazil. For example, between 2000 and 2017, Black women, women with low education, and Indigenous women had a higher risk of maternal death; the North and Northeast regions had the highest maternal mortality rates, while the South and Southeast regions had better health care [32]. More recent data from a 2024 study compared MMR rates among women of different races between 2017 and 2022. The overall MMR was 68 per 100,000 live births, but it was almost double among Black women compared to white women, especially in the Southeast [17]. From 2018 to 2023, an ecological study demonstrated that Indigenous women and youth faced intermunicipal travel for medical care, often exceeding 500 km, and these cases presented an MMR of up to 772.5 per 100,000 live births during the pandemic, highlighting territorial vulnerability [33].

Before COVID-19, Brazil was on a trajectory of reducing maternal mortality, approximately 57 per 100,000 live births in 2019, with a subsequent increase to approximately 67 in 2020 due to the pandemic [34,35]. Studies using time series models and spatial analysis (2010–2019) observed an increasing trend in late mortality (after 42 days), especially in the North, Northeast, and Central-West regions, with an average annual increase of up to 26% in late mortality in the Central-West [36].

This study presents some limitations inherent to the use of secondary data from official information systems. The first concerns the quality and completeness of maternal mortality records in the SIM, especially in regions with lower coverage and poor infrastructure, which can lead to underreporting or incorrect classification of causes of death. Incomplete or delayed investigations may distort the estimates of the MMR, masking regional or temporal disparities. This study presents some limitations inherent to the use of secondary data from official information systems. The first concerns the regional heterogeneity in data completeness and underreporting, particularly in the North and Northeast regions, where surveillance infrastructure and investigation coverage remain fragile. These differences may generate systematic biases in the distribution of maternal deaths, leading to the underestimation of rates in areas with lower data quality and the apparent overrepresentation of deaths in regions with more robust reporting systems. Moreover, the lack of uniformity in death investigations compromises data comparability and limits the capacity to evaluate the true effectiveness of regional health interventions.

Additionally, the absence of temporal trend modeling restricts the study’s ability to identify potential inflection points or significant shifts in maternal mortality over the two-decade period. The use of descriptive approaches allows for a clear overview of patterns but limits inferential interpretation. Future studies employing joint point regression, Prais–Winsten, or segmented linear models are recommended to detect temporal changes more precisely. Finally, as this is an ecological and retrospective study, it is not possible to establish causal relationships between sociodemographic, clinical, and regional variables and maternal mortality outcomes. The lack of multivariate modeling may also result in residual confounding, as independent effects of race, education, or geography could not be isolated.

The study offers important contributions to the field of maternal health by conducting a comprehensive analysis of maternal mortality in Brazil over two decades, based on a large volume of population data. The research allows us to identify temporal trends, regional patterns and social inequalities associated with maternal deaths, providing support for the planning of more equitable and territorially oriented public policies. The results also contribute to the monitoring of the SDGs, especially SDG 3 (Good Health and Well-Being) and SDG 10 (Reduced Inequalities), in addition to promoting reflections on addressing racism in health services.

The implementation of ongoing training and continuing education programs for health care professionals is recommended, especially in regions with the highest incidence of maternal deaths. These actions should include training in the management of obstetric emergencies, early identification of risk factors, and ensuring care pathways based on good clinical practices. Strategies such as the use of realistic simulation, regional workshops, and integration with Ministry of Health guidelines (such as those of the Rede Cegonha) can significantly contribute to reducing preventable mortality.

## 5. Conclusions

Maternal mortality in Brazil remains a serious and preventable public health problem rooted in structural inequalities and weaknesses in obstetric care. The findings highlight the urgent need to strengthen maternal health networks and ensure qualified, continuous, and equitable care during pregnancy, childbirth, and the postpartum period. Strengthening primary health care, expanding access to prenatal and postpartum follow-up, and improving the integration between levels of care are crucial to preventing avoidable deaths.

For nursing and midwifery practice, this study underscores the strategic role of these professionals in identifying early warning signs, managing obstetric emergencies, and ensuring continuity of care after discharge. Continuous education, evidence-based protocols, and interprofessional teamwork should be prioritized to improve clinical outcomes and patient safety. Promoting reproductive justice in Brazil requires not only technical improvements but also ethical and political commitments to address racial, social, and territorial inequities. Investing in the qualification of health teams, surveillance systems, and user-centered care models is essential to advance toward the SDGs and guarantee every woman the right to safe and respectful motherhood.

## Figures and Tables

**Figure 1 nursrep-15-00396-f001:**
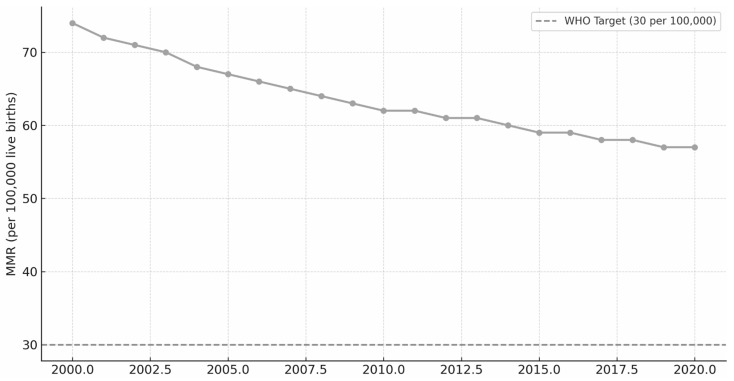
Temporal trend of MMR per 100,000 live births in Brazil, 2000–2020.

**Figure 2 nursrep-15-00396-f002:**
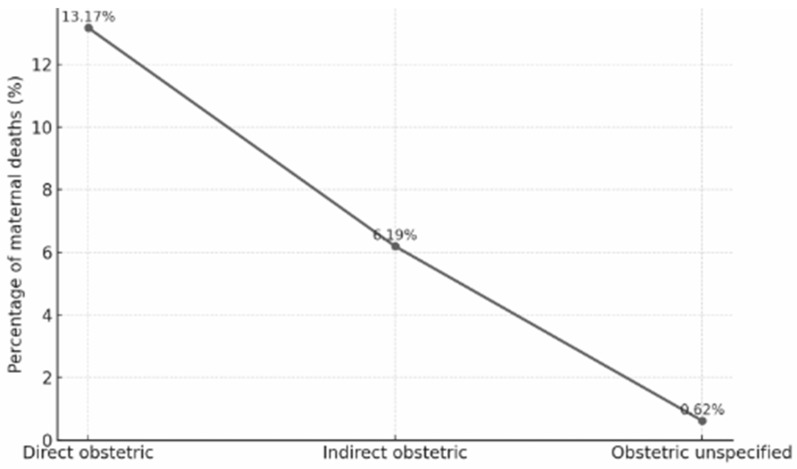
Distribution of maternal deaths by type of obstetric cause in Brazil, 2000–2020. Data from the SIM. Percentages represent the proportion of total maternal deaths classified as direct, indirect, or unspecified causes.

**Figure 3 nursrep-15-00396-f003:**
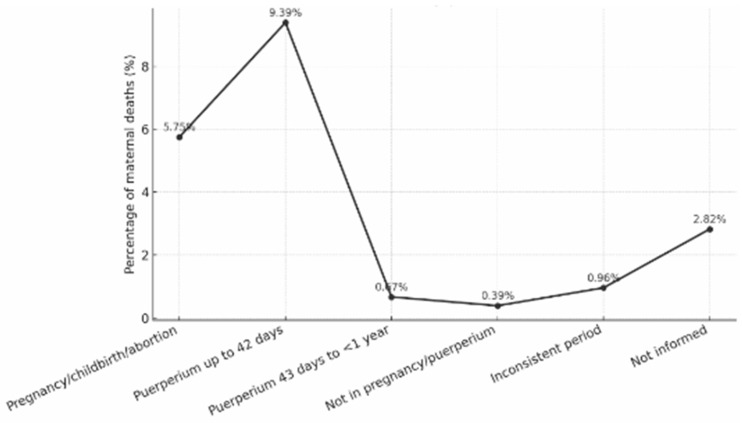
Distribution of maternal deaths by period of occurrence (pregnancy, childbirth/abortion, early or late puerperium) in Brazil, 2000–2020.

**Table 1 nursrep-15-00396-t001:** Sociodemographic distribution of maternal deaths in Brazil by region, age group, skin color, marital status and education, 2000–2020 (n = 40,907).

Category	Maternal Deaths (n=)	Maternal Deaths (%)
**Region**		
Northern Region	5147	12.6%
Northeast Region	13,748	33.6%
Southeast Region	14,129	34.5%
Southern Region	4555	11.1%
Central-West Region	3328	8.1%
**Age**		
10 to 14 years old	346	0.8%
15 to 19 years old	5105	12.4%
20 to 29 years old	16,381	40%
30 to 39 years old	15,548	38%
40 to 49 years old	3418	8.3%
50 to 59 years old	85	0.2%
70 to 79 years old	1	0%
Age ignored	23	0.1%
**Race**		
White	13,648	33.3%
Black	4446	10.8%
Yellow	123	0.3%
Brown	19,773	48.3%
Indigenous	566	1.3%
Ignored	2351	5.7%
**Marital status**		
Single	20,579	50.3%
Married	12,338	30.1%
Widower	307	0.7%
Legally separated	696	1.7%
Other	3819	9.3%
Ignored	3168	7.7%
**Education**		
None	1532	3.7%
1 to 3 years	4660	11.3%
4 to 7 years	9538	23.3%
8 to 11 years	12,151	29.7%
9 to 11 years	3	0.1%
12 years and over	3725	9.1%
Ignored	9298	23.7%
**Total**	40,907	100%

**Table 2 nursrep-15-00396-t002:** Distribution of maternal deaths according to ICD-10 chapter, type of cause, period, place of occurrence and investigation status—Brazil, 2000–2020 (n = 40,907).

Category	Maternal Deaths (n=)	Maternal Deaths (%)	IC 95%
**Chapter ICD-10**			
I. Infectious diseases	844	0.41%	0.38–0.44
II. Neoplasms	10	0.4%	0.002–0.008
IV. Endocrine/metabolic	1	0.5%	-
V. Mental disorders	36	0.1%	0.012–0.023
XV. Pregnancy/childbirth/puerperium	40,016	19.5%	19.393–19.737
**Type of cause**			
Direct obstetric	26,957	13.1%	13.033–13.327
Indirect obstetric	12,665	6.1%	6.088–6.297
Obstetric unspecified	1281	0.6%	0.592–0.661
**Períod of death**			
Pregnancy/childbirth/abortion	11,763	5.7%	5.650–5.852
Puerperium up to 42 days	19,206	9.3%	9.264–9.517
Puerperium 43 days to <1 year	1382	0.6%	0.640–0.711
Not in pregnancy or puerperium	803	0.3%	0.366–0.420
Inconsistent period	1976	0.9%	0.924–1.009
Not informed	5777	2.8%	2.753–2.896
**Place of occurrence**			
Hospital	37,288	18.2%	18.064–18.398
Other establishments	882	0.4%	0.403–0.460
Domicile	1513	0.7%	0.703–0.777
Public road	525	0.2%	0.235–0.279
Others	648	0.3%	0.292–0.341
Ignored	51	0.2%	0.018–0.032
**Death investigated**			
Investigated with record	23,123	11.3%	11.168–11.443
Investigated without a record	3548	1.7%	1.678–1.791
Not investigated	4482	2.1%	2.128–2.255
Not applicable	9754	4.7%	4.677–4.861

## Data Availability

The data presented in this study are available at Sistema de Informação de Mortalidade (https://opendatasus.saude.gov.br/dataset/sim).

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
