# Peer review of "Panorama of Two Decades of Maternal Deaths in Brazil: Retrospective Ecological Time Series"

_nursrep, 2025, doi:10.3390/nursrep15110396_

Round 1
Reviewer 1 Report (Previous Reviewer 3)
Comments and Suggestions for Authors
The improvements are visible compared to the older version, but minor corrections are needed.
Row 156-158: Repetitive information, please delete.
Row 188: Please correct the punctuation ”:” instead of ”;” .
Row 189-190: Lower case instead of upper case – ”place of death”, ”investigation status”.
Row 202: Please explain the abbreviation ”CSV”.
Row 211: Change the font to match the rest of the text.
Row 278: SDGs- please explain the abbreviation
Row 350: same as row 278
In the result section perhaps it would be more suitable some visualization of the data presented, some graphical representation: causes of death with separate representation during pregnancy, childbirth and recent early or late puerperium.
Author Response
Dear reviewer,
Thank you for your constructive comments and suggestions. All recommendations were carefully analyzed and incorporated into the manuscript as described below: 1) Repetitive information was removed to avoid redundancy and make the text more concise; 2) Punctuation was corrected, replacing the semicolon (;) with a colon (:), as suggested; 3) The words "Place of Death" and "Investigation Status" were adjusted to lowercase, maintaining consistency with the rest of the text; 4) The acronym "CSV" was explained when first mentioned, ensuring clarity for the reader; 5) The abbreviation "ODS" was properly explained when first mentioned and kept consistent in subsequent mentions; 6) The font was standardized to match the manuscript's standard; and 7) Following your suggestion, graphic representations were included illustrating the distribution of causes of maternal death by gestational period (pregnancy, childbirth, and early/early/late postpartum period), improving the clarity and interpretation of the findings.
Once again, we thank you for your valuable contributions, which certainly improved the quality of our work. Thank you.
Reviewer 2 Report (New Reviewer)
Comments and Suggestions for Authors
General Comments
The manuscript provides a comprehensive national overview of maternal mortality in Brazil over two decades. The use of national mortality and live birth data makes it a valuable contribution. The topic is timely and aligns with the Sustainable Development Goals and nursing research priorities. However, the analytical framework and manuscript structure need refinement for scientific precision.
Specific Comments
Title and Abstract
-
Include numeric summary of maternal mortality trends (e.g., MMR in 2000 vs 2020).
-
Make the conclusion more specific to maternal health system improvement.
Introduction
-
Reduce length; avoid repetitive discussion of inequalities.
-
Add a clear knowledge gap and justify why this national-level study is necessary now.
-
Emphasize how findings can inform nursing and midwifery interventions.
Methods
-
Replace “retrospective population-based cohort” with “retrospective ecological time-series analysis” for accuracy.
-
Explain temporal analysis (e.g., annual MMR calculation and statistical test for trends).
-
Clarify management of missing or unknown data (education, race).
-
Provide reference years and version numbers for R and used packages.
Results
-
Add a figure showing temporal trends in MMR by region.
-
Move large tables to supplementary materials; summarize key insights in text.
-
Avoid interpretation in the Results section—reserve for Discussion.
Discussion
-
Streamline and group by theme: Regional Inequalities, Obstetric Causes, Health System Factors, Nursing Implications.
-
Strengthen nursing and policy implications—describe how findings can inform workforce training or surveillance systems.
-
The inclusion of transgender and non-binary populations is commendable; however, integrate it with a local reference or policy framework.
-
Reduce repetition from Introduction.
Limitations
-
Add that the absence of multivariate or trend modeling limits causal inference.
-
Note that underreporting and incomplete investigations could bias MMR estimates.
Conclusions
-
Condense and emphasize practical implications for improving obstetric care and nursing practice.
-
Avoid repeating descriptive results.
References
-
Ensure consistency in numeric style.
-
Check that all DOIs are active and properly formatted.
3. Ethical and Reporting Standards
-
Ethical compliance and STROBE adherence are properly stated.
-
Consider mentioning data reproducibility or providing supplementary R code if journal policy allows.
4. Final Recommendation
Major Revision
Summary of Required Revisions
-
Clarify study design terminology and analysis type.
-
Add visual representation of MMR trends.
-
Simplify and restructure Introduction and Discussion.
-
Enhance nursing and policy relevance in Discussion and Conclusion.
-
Revise English language and ensure MDPI reference formatting compliance.
Author Response
Dear,
We are extremely grateful for your detailed observations and constructive contributions. Your suggestions were carefully analyzed and incorporated into the manuscript, as described below: 1) a numerical summary of maternal mortality trends was included, presenting the Maternal Mortality Rate (MMR) in 2000 and 2020, as requested. The conclusion was revised to be more specific, highlighting practical implications for improving the maternal health system and improving obstetric and nursing care; 2) the introductory text was shortened, removing repetitive passages on regional and sociodemographic inequalities. A clear knowledge gap was added, highlighting the scarcity of recent national analyses on temporal trends in maternal mortality in the post-pandemic context. The study's relevance was reinforced, justifying its current need and highlighting how its findings can inform nursing and midwifery interventions, focusing on surveillance, care, and professional training; 3) The term “population-based retrospective cohort” was replaced by “retrospective ecological time series analysis,” as recommended for greater methodological precision. The temporal analysis was detailed, including a description of the annual MMR calculation and the statistical test used to assess trends. The strategy adopted to deal with missing or unknown data (such as education level and race/color) was clarified. The versions of the R software and the packages used in the analysis were included; 4) An additional figure was included to represent the temporal trends of MMR by region of Brazil; 5) The section was reorganized by theme, as suggested: Regional Inequalities, Obstetric Causes, Health System Factors, and Nursing Implications. Policy and nursing implications were strengthened, with an emphasis on how evidence can inform public policy, professional training, and maternal health surveillance. The inclusion of transgender and nonbinary populations was maintained, now anchored in national references and policy frameworks. Previous repetitions in the Introduction were removed to optimize text flow. Observations were added regarding the lack of multivariate modeling and limitations regarding causal inference. It was acknowledged that underreporting and incomplete investigations can affect MMR estimates, strengthening methodological transparency; and 6) All references were reviewed to ensure consistency in numerical style and compliance with MDPI standards.
Once again, we deeply appreciate the recommendations, which contributed significantly to the scientific, methodological, and editorial improvement of the manuscript.
Round 2
Reviewer 2 Report (New Reviewer)
Comments and Suggestions for Authors
The manuscript titled “Panorama of Two Decades of Maternal Deaths in Brazil: Retrospective Ecological Time-Series” provides a valuable overview of maternal mortality trends and determinants in Brazil using secondary national data. The topic is highly relevant to the journal’s readership and aligns with current global efforts to reduce maternal deaths and achieve SDG 3.1.
The manuscript demonstrates strong public health importance and appropriate data use; however, several major revisions are required to strengthen its methodological transparency, analytical depth, and interpretive precision.
Major Comments
Originality and Contribution
The paper covers a significant period (2000–2020), yet the contribution beyond prior national surveillance reports remains unclear. Please highlight the novelty—what gap does this study fill compared to previous analyses by the Ministry of Health or Rede Cegonha?
Methods and Data Analysis
The term “retrospective ecological time-series” is used, but the analysis is primarily descriptive. Consider applying time-trend models (e.g., joinpoint regression, Prais–Winsten, or segmented linear regression) or justify their exclusion.
The absence of multivariate modeling should be discussed as a limitation with possible consequences for confounding effects (e.g., regional disparities independent of race or education).
Clarify procedures for data validation, exclusion of duplicates, and treatment of missing or “ignored” categories.
Results
Tables are extensive and informative but could be summarized to improve readability. Verify consistency between absolute numbers and percentages.
Figures 1 and 2 should include clearer legends and units. Consider adding a trend line figure showing the annual maternal mortality ratio over two decades.
Discussion
The discussion presents comprehensive contextualization but is at times overly descriptive. Refocus on linking results to specific public health actions—Rede Cegonha, PNAISM, and surveillance committee performance.
The inclusion of LGBTQIA+ health is conceptually valid but should be presented as an implication for future policy and data inclusion since such variables are not captured in the current dataset.
Limitations
Expand on regional underreporting and data completeness, emphasizing potential bias from differences in surveillance capacity.
Note that lack of trend modeling restricts the ability to identify temporal inflection points.
Language and Style
The manuscript is readable but would benefit from English proofreading to enhance flow and remove redundancy.
Use concise phrasing in the Results and Methods sections.
Minor Comments
Abstract: Add data sources (SIM, SINASC) and type of statistical tests used.
Introduction: Include the latest WHO global MMR for comparative context.
Tables: Simplify and unify decimal precision.
References: Ensure consistent MDPI formatting and DOI inclusion for all references.
Ethical Statement: Reconfirm that data are anonymized and publicly available.
Conclusion: Strengthen the final paragraph with actionable recommendations for maternal surveillance, health workforce training, and nursing leadership roles.
Overall Recommendation
The study is relevant and well-grounded but requires substantial revision in analysis and focus before acceptance.
Recommendation: Major Revision
Author Response
We sincerely appreciate the detailed and constructive feedback on our manuscript. Each comment was carefully addressed, and substantial revisions were made to improve the study's clarity, analytical rigor, and public health relevance. Below, we provide a detailed, point-by-point response:
1) We revised the end of the Introduction to clarify the originality of our study. Specifically, we emphasize that, unlike previous reports from the Ministry of Health and the Stork Network, this research integrates two decades of population data and stratifies maternal deaths by sociodemographic, racial, and regional dimensions. This fills a gap by linking the temporal and structural determinants of maternal mortality through a comprehensive national panorama
2) We justify the exclusion of regression-based models (joinpoint, Prais-Winsten, segmented regression) at the end of the Statistical Analysis section, explaining that data heterogeneity and the lack of annual strata between regions limited model comparability. Therefore, the focus was descriptive, ensuring methodological transparency
3) We have included a new subsection in Data Sources and Variables detailing the steps for data validation, duplicate removal, and missing value handling, with sensitivity analyses to check consistency
4) Tables 1 and 2 present relevant sociodemographic and clinical variables. Percentages have been recalculated and rounded to one decimal place for consistency. The captions for Figures 2 and 3 have been rewritten for clarity, including data sources and explanations of percentages. Additionally, a new Figure 1 has been created, illustrating the time trend of the maternal mortality ratio (MMR) from 2000 to 2020, using data from the SIM
5) We restructured the Discussion to explicitly connect the findings to the Stork Network, the National Policy for Comprehensive Women's Health Care, and the Maternal Mortality Committees, emphasizing how regional inequalities and underreporting impact the effectiveness of these programs. We modified the paragraph to clarify that LGBTQIA+ inclusion is a future policy implication, given that gender identity variables are not captured in the current SIM dataset. Relevant references were retained and contextualized as recommendations for future research
6) We expanded the limitations section to discuss regional differences in data completeness, particularly in the North and Northeast regions, and their potential biases. The text now addresses how surveillance capacity influences data accuracy and comparability
We thank the reviewer for recognizing the relevance of this study. The manuscript was carefully reviewed to improve analytical accuracy, contextual relevance, and alignment with journal standards. We believe the revised version now fully addresses all concerns and offers a more focused and policy-oriented contribution to maternal health research in Brazil.
This manuscript is a resubmission of an earlier submission. The following is a list of the peer review reports and author responses from that submission.
Round 1
Reviewer 1 Report
Comments and Suggestions for Authors
The study offers an extensive amount of data that reflects changes in maternal health, with 40,907 maternal deaths over 20 years. The study's results are enhanced by its accurate and well-defined methodology, which adheres to the STROBE requirements and incorporates in-depth statistical analysis. The paper highlights significant concerns, including racial and geographical disparities, and aligns with the United Nations' Sustainable Development Goals (SDGs). The recommendation can be more specific in improving the maternal care by mentioning a training program for healthcare providers in high-risk areas.
Some reference names were written in the Brazilian language, which is difficult to understand.
Some references were very old, dating back 18 years. Update the references.
Comments on the Quality of English Language
It's good, but there were minor grammatical errors.
Author Response
Dear reviewer,
We sincerely thank you for your comments and careful reading of our manuscript. We consider your observations extremely valuable for improving the work. Below, we respond point by point to the suggestions presented:
1. We fully agree with the importance of presenting more applicable recommendations. Therefore, we have added a specific suggestion at the end of the discussion related to the training of health professionals through continuing education programs, focusing on the most vulnerable regions and populations. This recommendation aims to directly contribute to reducing the observed disparities and is aligned with the Sustainable Development Goals.
2. We reviewed all references and adjusted the authors' names that were spelled in Portuguese, standardizing them in accordance with international standards, to ensure clarity and overall comprehension of the manuscript. We emphasize, however, that the names of entities and representative bodies remained with their original names.
3. We carefully reviewed the bibliography and replaced older references with more recent studies published in the last 10 years, whenever possible. However, we have retained some classic references, which are fundamental to the theoretical foundation of the topic, justifying their retention in the new version.
4. We have corrected the minor errors identified to ensure greater accuracy and fluidity in the text.
We are available for any further clarification and hope that the changes made meet your expectations. Thank you.
Reviewer 2 Report
Comments and Suggestions for Authors
Dear authors, after reviewing the proposed manuscript, I detail below certain points that should be reviewed. The results provided highlight a problem present in many underdeveloped and developing countries, and as the data show also in Brazil. National and international organizations must pay attention to this type of evidence and bet on the prevention of the factors that contribute to this type of death.
Line 67_ Introduce the objective of the study at the end of the introduction, including, if appropriate, the respective study hypothesis.
Line 75-81: Not part of the subject to be studied, it can be deleted.
Line 139-140: quote
Line 150: It is suggested that the manuscript be carefully reviewed to conform to the specified results presentation standards.
Line 153-156: Include in point 2.2
Line 161: Clarify if these are the eligibility criteria.
Line 167-168: quote
Line 208-211: Specify the item to which they refer in order to access the data
Line 213: review the percentages presented, since it is convenient to describe the relative percentages for each of the variables explored, not for the total sample.
Line 276-277: It would be useful to provide data on pregnancies in these groups of women in order to reach enlightening conclusions.
Line 382-383: Provide data on the population in these areas
Line 396: cite "Red de Cigüeñas".
Line 411-412: quote
Line 430-441: It could be deleted because it does not proceed in this discussion
Line 443: An important limitation is the lack of information on the relative percentage of pregnancies and births in these population groups where maternal death seems to be more frequent. For example, it is normal that there is a higher frequency of deaths in hospital, since most births and postpartums are carried out in institutions.
Citation style: what rules does it meet?, I observe in the text that when citing the authors the year is entered in parentheses and the citation at the end of the sentence, it is recommended to introduce the citation immediately after the author's name if it is cited in the text, and to enter the year only ́ if it is relevant.
Author Response
Dear reviewer,
We greatly appreciate your careful reading and valuable contributions to our manuscript. Based on your observations, we have made the necessary changes, as detailed below:
1. We have added, at the end of the introduction, the objective of the study in a clear and direct manner, also including the hypothesis that guided the investigation, as suggested.
2. We agree with the observation and have excluded the aforementioned excerpt because it is not directly related to the central theme of the study.
3. The appropriate references have been inserted in these excerpts, ensuring the statements are adequately substantiated.
4. We have carefully revised the results section to ensure compliance with scientific presentation standards, organizing the data in a clearer, more precise, and objective manner.
5. The suggested content has been incorporated into item 2.2 (methodological procedures), adjusting the text structure as recommended.
6. We corrected the presentation of the data, using percentages relative to each variable analyzed, rather than the total sample, as previously presented.
7. We acknowledge the relevance of this suggestion. However, we would like to point out that data on the number of pregnancies are not available in the databases used. This limitation was acknowledged and discussed in the study limitations section.
8. We included population data for the highlighted areas to better contextualize the findings and reinforce the interpretations presented.
9. We added this important limitation at the end of the discussion, emphasizing that the lack of data on the total number of pregnancies by population subgroup may influence the interpretation of the findings. We also clarify that, since most births in Brazil occur in hospital settings, it is expected that most deaths will also be recorded in these institutions.
10. We standardized the citation style throughout the manuscript, respecting the journal's standards.
We thank you once again for your careful reading and for your contributions, which certainly enriched our work. Thank you
Reviewer 3 Report
Comments and Suggestions for Authors
The authors present an extensive retrospective population-based study conducted in a 20-years period, focusing on maternal mortality in Brazil. They systematically analyzed data regarding the causes of maternal deaths during this time-frame, incorporating concepts such as near-miss mortality in the introduction section – a critical indicator assessing the effectiveness of healthcare systems. The study examines socio-demographic characteristics, geographical disparities, and classifies causes of death into three main categories: direct obstetrics, indirect obstetrics and unspecified causes. The main conclusion was that according to ICD-10 classification system, most deaths were related to pathologies described and coded into chapter XV. The article contains essential information and can be the start up point for future healthcare management strategies and improvements.
It needs some clarifications before publication.
Why did you mentioned near missed mortality in the introduction section if the rest of the article is concentrated on something else?
“With 40,016 cases (19.56% of the analyzed base, considering technical proportionality by total causes listed)” – please rephrase for clarity – how many deaths in total in Brazilian women in the 20-years that were analyzed, and how many from those were maternal deaths.
Rows 174-175 – “Period of death: during pregnancy, childbirth or abortion; puerperium (up to 42 days and from 43 days to less than 1 year) – please rephrase for clarity
Rows 246-280 – repetitive information – see the first paragraphs from the results section. Please rephrase and rearrange them
“Source: Prepared by the authors. Brazil” – I suggest to delete this from bellow tables and figures
Rows 326-360 – repetitive information – Please rephrase and rearrange them
Discussion section – please comment more your results compared to those in literature especially in different other regions than yours.
Author Response
Dear Reviewer,
We deeply appreciate your careful reading and the thoughtful comments you made to our manuscript. Your contributions were essential to improving the clarity, cohesion, and scientific quality of the work. Below, we respond in detail to each of your suggestions:
1. Thank you for this important observation. Initially, the study envisaged a more in-depth analysis of maternal mortality due to near miss, but as the review progressed, we chose to focus exclusively on causes of maternal death coded by ICD-10. Therefore, we removed the section referring to near miss from the introduction to maintain consistency between the objectives, methodology, and results presented.
2. We rewrote the paragraph to make it clearer and more direct. We now explicitly report the total number of deaths of women of childbearing age recorded in the 20 years analyzed and, within this total, how many were classified as maternal deaths according to the study criteria.
3. We reorganized the initial paragraphs of the results section to avoid repetition and improve readability. The data presentation was condensed, grouping similar information and eliminating redundancies.
4. We followed the suggestion and removed this expression from the table and figure captions, retaining only relevant information and avoiding unnecessary repetition.
5. We rewrote and reordered this discussion section to eliminate repetition and make the text more cohesive, emphasizing the interpretation of the findings.
6. We expanded the discussion by incorporating international and regional studies that analyze maternal mortality in countries with distinct sociodemographic profiles. Recent references have been included to allow for a critical comparison of the findings, highlighting similarities and differences regarding the predominant causes, geographic inequalities, and the impact of public policies on maternal mortality.
Once again, we appreciate your contributions, which certainly made our manuscript more robust and relevant. We remain at your disposal for any further adjustments. Thank you.
Round 2
Reviewer 2 Report
Comments and Suggestions for Authors
Dear authors, after the second review, shortcomings have also been detected that need to be reviewed
Agenda for Sustainable Development Goals is not adequately cited in the text (line 128-129)
Line 169-171: explicitly describe what inclusion criteria are, since it seems that all three criteria must be met, i.e., specify that the women included must have met one of the criteria cited
Line 162 and 178: The abbreviation is placed on the first time it is cited, and it has not been referenced as requested in the first review.
Line 166: cite Chapter XV of ICD-10.
Line 162-175: Review Repeated Paragraphs
Line 189: repeated phrase
Line 178-191: Review punctuation marks in the description of the variables.
Line 221: Please, as requested in the first review, refer to public access to data by reference with an online link if possible. This aspect is very essential.
Line 204-209: This paragraph does not apply, it must be included in the description section of the variables
Line 211-218: in this section describe only the statistical methodology used; It is not appropriate to describe additional information on the methods, this can be done in the discussion section
Line 226; In general, during the results, it is recommended to provide data objectively, avoiding repeating the data provided in the text and tables
Table 1:
- Reviewing the age data, from the 50-59 range to 70-79, where one death was also recorded
- Review the education data, from the 8-11 range to 9-11, where there are only 3 cases
Figure 1:
- I recommend authors use a different type of chart.
- The data presented in Table 1 are repeated.
- Consider presenting some data in table format and others that you want to highlight in graph format.
- It is evident from the distribution of points that the erroneous data in Table 1 have been transferred to this graph
- The variable "ignored" must be specified for each variable, because as can be seen, 3 of the variables converge at the same point
Line 257-261: it is not necessary to repeat what is going to be exposed in the table or other information not strictly derived from the study, this is already defined in the methodology section
Table 2:
- It has not been properly revised, the results continue to be expressed in the form of absolute frequencies. Review and provide the frequencies relative to each independent variable.
- In the variable " the data relating to women who died in a period that does not correspond to the criteria defined in the method section are not understood
Figure 2: Same as described in Figure 1
I recommend that the authors pay attention to the above considerations, which once resolved will allow me to adequately review the discussion and conclusions sections.
Reviewer 3 Report
Comments and Suggestions for Authors
I am satisfied with the corrections made.